# Serum Antibody Titres Against *Porphyromonas gingivalis* Fim A in Patients with Periodontitis with and Without Diabetes Mellitus Type 2

**DOI:** 10.3390/ijms26104726

**Published:** 2025-05-15

**Authors:** Sabine Groeger, Nathalie Mueller, Jens Martin Herrmann, Joerg Meyle

**Affiliations:** 1Department of Periodontology, Dental School, Justus-Liebig-University of Giessen, 35392 Giessen, Germanyprof.dr.j.meyle@t-online.de (J.M.); 2Department of Orthodontics, Dental School, Justus-Liebig-University of Giessen, 35392 Giessen, Germany

**Keywords:** *P. gingivalis* FimA, antibody titres, ELISA, periodontitis, IgG

## Abstract

Periodontitis and type 2 diabetes mellitus are interconnected in a bidirectional relationship, yet the underlying mechanisms remain poorly understood. *Porphyromonas gingivalis* (*P. gingivalis*), a key periodontal pathogen, has been implicated in both conditions. This study investigates the association between antibody responses to *P. gingivalis* and the coexistence of periodontitis and diabetes, aiming to explore its potential as a biomarker for early screening. The developed enzyme linked immunosorbent assay (ELISA) was used to analyse in a pilot study the serum of subjects with periodontitis (*n* = 26), subjects with periodontitis and type 2 diabetes mellitus (*n* = 15), and healthy individuals (*n* = 13) for immunoglobulin (Ig)G- and IgM-antibody titres against FimA. A statistically significant difference (*p* < 0.001) between the IgG titres of the three study groups was observed. Patients with periodontitis and type 2 diabetes mellitus showed the highest IgG titres and thus the highest level of IgG antibodies, followed by periodontitis patients and finally the orally healthy subjects. This study demonstrated the establishment of a reliable and reproducible ELISA for the detection of antibodies against FimA from *P. gingivalis* W83. In addition, the outcomes of this study can be used to develop a screening assay that can indicate existing periodontitis and type 2 diabetes mellitus based on IgG antibody titres against FimA from *P. gingivalis*.

## 1. Introduction

Periodontitis is an inflammatory, multifactorial disease of the periodontium that is initiated by bacteria [1,2]. The development of periodontitis is caused by a dysbiosis in the oral microbiome, which can be initiated by periodontal bacteria and the subsequent immune response of the host [3]. As a result of this imbalance, inflammatory processes cause reversible gingivitis [4]. If gingivitis is not cured, the further course of the reaction of the innate and acquired immune system leads to the destruction of the periodontium and thus to periodontitis.

*Porphyromonas gingivalis* (*P. gingivalis*) is regarded as main cause of periodontitis and thus brings this pathogen into focus for possible new treatment approaches [5]. Depending on the bacterial strain, it exhibits different virulence factors that determine its pathogenic properties [6]. They include cysteine proteases (gingipains), hemagglutinin, hemolysin, a capsule, lipopolysaccharides and fimbriae [7,8,9]. Fimbriae are filamentous structures anchored in the outer membrane of the cell surface. They play a crucial role in co-aggregation, the colonisation of host cells, biofilm formation and the development of oral dysbiosis. Thus, fimbriae are of utmost importance in the interaction with other bacteria and host cells.

Fimbriae nomenclature differentiates the long fimbriae, called FimA, and the short fimbriae, named Mfa fimbriae. FimA takes part in mediation of the binding to Gram-positive and Gram-negative bacteria, including oral streptococci and Actinomyces species (such as *Actinomyces viscosus* (*A. viscosus*), *Treponema denticola* (*T. denticola*), *Streptococcus gordonii* (*S. gordonii*) and *Streptococcus oralis* (*S. oralis*). FimA therefore plays a role in biofilm formation [7]. It is involved in binding to human epithelial cells, endothelial cells, spleen cells and peripheral monocytes and the subsequent release of inflammatory cytokines such as tumour necrosis factor-α (TNF-α), interleukin (IL)-1β, IL-8 and IL-6. FimA also binds cell adhesion molecules such as intercellular adhesion molecule (ICAM)-1, vascular cell adhesion protein (VCAM)-1 and P- and E-selectin [7,10,11]. FimA is one of the main virulence factors of *P. gingivalis*. It is of great importance for the adhesion and invasion of the bacterium. It is able to subvert the immune system, thus helping the bacterium to survive and multiply even within cells. Currently in *P. gingivalis*, the FimA genotypes I, II, III, IV and V have been classified and are associated with its virulence [7,11].

Diabetes mellitus type 2 is a metabolic disease that is based on insulin resistance in the body’s own cells, causing the β-cells of the pancreas to overproduce insulin to compensate, which cannot be maintained permanently. As a result, saccharides cannot be absorbed into the cells and increased blood sugar levels occur. As further consequences, proteins in the serum are coupled with saccharides non-enzymatically through glycation. These products are called AGEs (advanced glycation end products) [12]. These AGEs are not metabolised and can accumulate in blood vessels, where they cause damage. The haemoglobin in the blood can also be glycated and provides a method of blood sugar control over a period of approx. 90 days [13]. The HbA1c value is determined for this purpose. The level of haemoglobin that has reacted with saccharides correlates with it and indicates inadequate blood sugar control. A pathological HbA1c value starts at 6.3% and higher [14]. Cells of the innate immune system, such as macrophages and monocytes, have receptors, so-called RAGEs (receptors for AGE), which can bind AGEs. This binding leads to the release of reactive oxygen molecules, the activation of transcription factors such as nuclear factor (NF)-kB and the secretion of pro-inflammatory cytokines, which initiate defence processes and inflammatory reactions [15].

Associations of *P. gingivalis* with cardiovascular diseases, type 2 diabetes mellitus and premature birth of infants have already been shown [7]. Furthermore, connections between diabetes mellitus and periodontitis have been uncovered [16,17]. Studies show a bidirectional relationship between the two diseases [18,19]. In both diseases the cells of the innate host response play an important role and release pro-inflammatory cytokines such as IL-1β, IL-6 and TNF-α after activation. Previously published studies have indicated associations between antibody titres against *P. gingivalis* and periodontitis, as well as diabetes mellitus [20,21].

For monitoring individual infections, the determination of specific antibody titres is very promising and helpful. Respective assays should further discriminate between patients with periodontitis and healthy individuals, since *P. gingivalis* has been detected not only in patients with periodontitis but also in healthy subjects [22,23].

Therefore, the aim of this study was to develop an enzyme linked immunosorbent assay (ELISA) that detects antibodies against FimA from *P. gingivalis* W83. Subsequently, sera from patients with periodontitis, patients with periodontitis and type 2 diabetes mellitus and healthy subjects were tested for immunoglobulin (Ig)G and IgM antibodies against *P. gingivalis* W83 FimA.

## 2. Results

An antiserum with the rabbit polyclonal antibodies was tested in comparison to the preimmune rabbit serum. The evaluation showed that specific bonds could be detected. The optical density (OD) values of the antiserum were four times higher per dilution level than the absorbance values of the preimmune serum. The titre of the antiserum was at a dilution of 1:200 (Figure 1).

To determine the optimal concentration for coating the individual wells, different concentrations of rFimA were tested using the same ELISA protocol. The antiserum and preimmune serum (negative control) from the same rabbit was used for the test. An increase in the amount of rFimA from 25 ng to 50 ng per well resulted in an increase in optical density of a factor of 1.7-fold. In comparison, an increase from 50 to 100 ng per well led to a 1.1-fold increase in the OD and a further increase from 100 ng to 150 ng per well only induced a 1.008-fold increase. The standard deviations of the optical densities at 50 ng rFimA per well as the determined mean value of the OD showed the best readout compared to the other tested concentrations (Figure 2).

Because of this, the concentration for the antigen coating was set at 50 ng rFimA per well. The optical densities of the tested monoclonal antibodies showed that specific bonds could be detected in the antibodies of several clones. The highest readout was found using the antibodies of clone no. 55, and the second highest was found using the antibodies of clone no. 189 (Figure 3).

For further investigations of human sera, the identical protocol for the ELISA as described above was used. In Table 1, the demographic data of the chosen sera from patients/controls are depicted.

The measured optical densities showed that the median IgG titres of the orally healthy individuals were at a dilution of 1:200, in the periodontitis patients at 1:800 and in the patients with periodontitis and diabetes mellitus type 2 at 1:3200. Table 2 shows the descriptive statistics and Table 3 shows the distribution of the frequencies of the detected titres. IgG titres in healthy subjects most frequently occurred at a dilution of 1:200 (30.77%) and 1:400 (30.77%); in periodontitis patients at a dilution of 1:1600 (26.92%); and in patients with periodontitis and diabetes mellitus type 2 at a dilution of 1:3200 (53.33%).

The Shapiro–Wilk test showed that the data were not normally distributed. The Kruskal–Wallis test (Table 4) followed by Dunn’s post hoc test with the Bonferroni–Holm correction (Table 5) showed that there was a significant difference between the IgG titres of healthy subjects and periodontitis patients (*p* = 0.008) and between the healthy subjects and patients with periodontitis and diabetes mellitus type 2 (*p* < 0.001). In addition, mean IgG titres of periodontitis patients and periodontitis patients with diabetes mellitus were also significantly different (*p* = 0.02) (Figure 4).

The ELISA test for the detection of IgM antibodies against FimA of *P. gingivalis* W83 showed a median IgM titre at a dilution of 1:100 in healthy subjects and a median dilution of 1:200 in the periodontitis patients. Table 6 shows the descriptive statistics and Table 7 the distribution of the frequencies of the detected titres.

The median dilution in the patients with type 2 diabetes mellitus and periodontitis was 1:200. The titre that occurred most frequently in the group of healthy subjects was the dilution 1:200 (30.77%); in the group of periodontitis patients, it was 1:50 (26.92%) and 1:400 (26.92%) (Table 8). In the group of patients with type 2 diabetes mellitus and periodontitis, the most frequently appearing dilution was 1:200 (46.67%). The highest dilution at which IgM antibodies were detected was 1:400 in the healthy subjects and in patients with periodontitis and diabetes mellitus type 2. The highest dilution was detected in the group of periodontitis patients at 1:800 (Table 7).

Employing the Shapiro–Wilk test (Table 9), the data were shown to be not normally distributed. The Kruskal–Wallis test showed no statistically significant difference in the IgM titres (Figure 5).

Calculation of Spearman’s rank correlation, revealed a significant relationship between the IgG titres and IgM titres. (*p* < 0.01) (Figure 6).

The positive relationship between the parameters was of medium strength with a Spearman rho of 0.421 (Table 8).

## 3. Materials and Methods

The inclusion and exclusion criteria for the chosen sera of patients/controls are shown in Table 9.

Recombinant Fim A was cloned in *Listeria innocua* and prepared and characterised as previously described (Groeger et al., 2021) [24].

As a first step, the optimal concentration of FimA as capturing antigen was determined. Therefore, concentrations from 10/50/150/200/250/300/350/400/450/500 ng/well were prepared in 100 mM HEPES Puffer and filled in Nunc MaxiSorp™ 96-Well-ELISA-microtiter plates (Thermo Fisher Scientific, Rheinfelden, Germany).

After incubation for 16 h (h) at 4 °C, the plates were washed 3 times with phosphate-buffered saline (PBS) (Greiner Bio-One GmbH, Frickenhausen, Germany) pH 7.4. Non-specific binding sites were blocked with PBS containing 1% non-fat dry milk 1 h at 37 °C.

Previously generated polyclonal rabbit antiserum against *P. gingivalis* W83 FimA (Davids Biotechnology, Regensburg, Germany) in dilutions of 1:25, 1:50, 1:100, 1:200, 1:400, 1:800, 1:1600 and 1:3200 was filled in the wells and incubated 2 h at 37 °C. Pre-immune serum from the same rabbit was used as negative control. Furthermore, also previously generated monoclonal antibodies were tested, for the binding activity of different clones. The antibodies were used in a concentration of 0.05 mg/mL and incubated for 2 h at 37 °C.

Detection was performed by application of horseradish peroxidase (HPR) conjugated anti-rabbit-Ig (Invitrogen, Thermo Fisher Scientific, Rheinfelden, Germany) after 3 washes with PBS for 1 h at 37 °C. Then, the plate was washed again 3 times, and a substrate (tetramethylbenzidine solution = TMB) (Thermo Fisher Scientific, Rheinfelden, Germany) was added for 20 min. The reaction was stopped by addition of 1 M HCL and the OD was measured in a Mithras LB 960 microplate multireader (Berthold Technologies, Bad Wildbad, Germany).

After the optimal antigen concentration for coating was determined, it was used as a base for further experiments. Plates were coated, washed and blocked as described above. Serum samples from 42 patients were taken, i.e., 26 patients with chronic periodontitis documented with a periodontal screening index (PSI) = 4; 16 patients with periodontitis and diabetes mellitus (PSI = 4, HbA1c > 8.5%); and 13 healthy individuals (PSI < 1). Sera were diluted 1:25, 1:50, 1:100, 1:200, 1:400, 1:800, 1:1600 and 1:3200 and placed in the wells. The dilution buffer was used as a negative control. Detection was performed by application of horseradish peroxidase (HPR) conjugated anti-human-IgG and IgM (Invitrogen, Thermo Fisher Scientific, Rheinfelden, Germany) after washing 3 times with PBS for 1 h at 37 °C. The plates were washed again 3 times, and the substrate was added 30 min. The reaction was stopped, and the ODs were measured in the multireader, as described above. The mean value of the blanks, which was determined from the OD values of the blanks, was subtracted from the optical densities determined for the respective dilution level.

Statistical analysis was carried out using Microsoft Excel and JASP 0.16.4.0 (University of Amsterdam, Amsterdam, The Netherlands) for data analysis. The collected data were determined for normal distribution using the Shapiro–Wilk test. A normal distribution was assumed for the age of the test subjects. The variance homogeneity test of the study groups was carried out using the Levene test [25].

The independence and comparability of the study groups were determined using the chi-square test of independence and one-way analysis of variance (ANOVA).

The Kruskal–Wallis test was used to check a statistically significant difference between the groups of oral healthy controls, subjects with periodontitis and type 2 diabetes mellitus in the recorded measurements (IgG and IgM titre). If the statistical significance of *p* < 0.05 was shown, Dunn’s post hoc test [26] was then carried out with Bonferroni–Holm corrections [27].

Associations between IgG and IgM titres were tested using Spearman’s rank correlation determination. The significance level for all tests was set at *p* < 0.05.

## 4. Discussion

The aim of this study was to develop an ELISA that is able to detect IgG and IgM antibodies against FimA from *P. gingivalis* (W83). An indirect ELISA was used for this purpose, since antibody-based detection methods are established procedures in most laboratories. Further advantages of the ELISA technique are that it achieves high sensitivity and specificity. Furthermore, it can be used to carry out efficient and simultaneous tests in the long term. It is also suitable for the detection of antigen-specific antibodies by coating the bottom of the wells of a microtiter plate with the respective antigen [28]. Recombinant FimA (rFimA) of *P. gingivalis* W83 was employed for antigen coating. This strain forms FimA type IV, a type that is frequently found in patients with periodontitis [29,30]. The used rFimA was purified and functionally classified before use in the ELISA and furthermore produced in the Gram-positive bacterium *L. innocua*, which prevented contamination during protein isolation with lipopolysaccharides [24]. In the present study, at first non-purified polyclonal rabbit antisera as well as monoclonal mouse antibodies were used to assess the optimal antigen concentration for coating. The advantage of polyclonal antibodies is that they recognise several epitopes on the antigen and can still bind to it, if an epitope has been lost due to fixation processes or partial denaturation. The use of monoclonal antibodies requires a higher sensitivity because these antibodies only bind a specific epitope of the antigen. Evan partial denaturation may lead to loss of specific binding. The advantages of monoclonal antibodies are that they are produced by hybridoma cells, which can produce a large amount of antibodies through cloning [31]. In addition, antibodies with the same specificity can be produced, enabling the good reproducibility of tests. Both approaches led to the finding of the optimal antigen concentration and a successful establishment of the developed ELISA.

The results of serum testing in healthy subjects, periodontitis patients and patients with periodontitis and diabetes mellitus type 2 for IgG antibodies against FimA from *P. gingivalis* W83 showed that the IgG titres of the three test groups differed significantly. The study group of healthy subjects showed the lowest titres and thus the fewest IgG antibodies. The titres of the patients with periodontitis were higher compared to the titres of the orally healthy subjects. The measured values indicate that periodontitis patients develop a stronger adaptive immune response to an infection with *P. gingivalis*, which leads to the formation of more IgG antibodies against one of the pathogen’s virulence factors. This observation is supported by the chronic nature of periodontitis, which, for multifactorial reasons, leads to the recurring exacerbations of the inflammatory immune processes in the periodontium [1]. These results compared with the titres of the group of patients suffering from periodontitis and diabetes mellitus type 2, demonstrate that in this group the highest titres were most frequently detected at a dilution of 1:3200. This leads us to the conclusion that diabetes mellitus type 2 disease in periodontitis patients induces an even higher increased inflammatory adaptive immune response, causing the enhanced formation of IgG antibodies against the *P. gingivalis* virulence factor FimA. The elevated blood sugar levels of diabetes mellitus patients and the formation of AGEs with their binding to RAGEs initiate defence processes and inflammatory reactions [15]. These processes can lead to tissue and organ damage and thus to increased systemic inflammation processes, which appear to exacerbate periodontitis. Previously published studies showed a bidirectional relationship between periodontitis and type 2 diabetes mellitus and thus support the results of the present study [18,19].

Studies showed that *P. gingivalis* has been detected in orally healthy subjects, but less frequently than in patients with periodontitis. Ingalagi et al., 2022 detected *P. gingivalis* in 53% of healthy subjects using a PCR test [22].

The transmission of *P. gingivalis* by family members could be a possibility for the high incidences of antibodies against the bacterium [32]. This theory could explain why healthy subjects develop antibodies against the bacterium even though periodontitis is not present. In addition, the smaller sample size of healthy subjects could be another reason for the results. Nevertheless, this study clearly demonstrates that healthy subjects develop fewer IgG antibodies against *P. gingivalis* FimA than periodontitis patients and those with diabetes mellitus type 2.

In various studies, over 5000 serum samples from subjects were evaluated. The results showed a strong connection between high anti-*P. gingivalis* antibody titres and diabetes mellitus as well as periodontitis [20,21]. Sims et al. showed that the median serum IgG titres against *P. gingivalis* were significantly higher in patients with type 1 diabetes mellitus and periodontitis than in patients without type 1 diabetes mellitus [33,34]. A possible reason for this was presented in the study by Montevecchi et al., 2021 [35]. Patients with type 2 diabetes mellitus and periodontitis were shown to exhibit a higher proportion of pathogenic bacteria, such as *P. gingivalis*, than patients with periodontitis but without diabetes [35]. In contrast to the studies which used whole bacterial species or complex antigenic components of *P. gingivalis*, in our study, a highly purified recombinant FimA antigen, one of the major virulence factors of *P. gingivalis*, was used. This approach offers the much higher stability and reproducibility of the established test system.

The testing of the sera of healthy subjects, periodontitis patients and patients with periodontitis and diabetes mellitus type 2 for IgM antibodies against FimA from *P. gingivalis* W83 showed a slight tendency towards higher IgM titres in the patient groups compared to healthy subjects. This difference did not show statistical significance. One possible cause could be the number of subjects, which could have influenced the statistical significance. In addition, IgM antibodies are formed as primary antibodies in an acute adaptive immune system to an antigen by the host since periodontitis is a chronic disease that is associated with different stages of inflammation and immune responses [36]. The average half-life of IgM class immunoglobulins is 5–10 days, so they can only be detected for a limited period during acute infection [37]. Another reason for the measured results could be that IgG immunoglobulins are predominant in the serum. They can interfere with the reaction between the IgM antibodies and the rFimA, since IgG antibodies have a higher affinity for their antigens and can thus prevent the binding of the IgM antibodies (Deshpande, 1996) [38].

The results of this study showed that increased serum IgG titres against FimA of *P. gingivalis* W83 may indicate periodontitis and type 2 diabetes mellitus. The early detection and treatment of periodontitis and diabetes mellitus play a crucial role in the health of a population. Further studies with larger sample sizes will provide greater statistical support for the results shown in this work. Furthermore, future research will provide pathophysiological evidence of associations between antibody titres, periodontitis, and diabetes mellitus. In the future, the newly developed test could be a useful approach for further studies on antibody titres in different areas. It holds great potential for observing therapeutic outcomes not only based on common clinical but also based on immunological parameters. 

## 5. Conclusions

A reliable and valid ELISA test for the detection of antibody titres against *P. gingivalis* FimA was established.

The results from the analysis of IgG antibodies against *P. gingivalis* FimA demonstrate that this virulence factor induces a clear humoral immune response.

In addition, increased IgG titres against *P. gingivalis* FimA could possibly indicate periodontitis in combination with diabetes mellitus type 2.

## Figures and Tables

**Figure 1 ijms-26-04726-f001:**
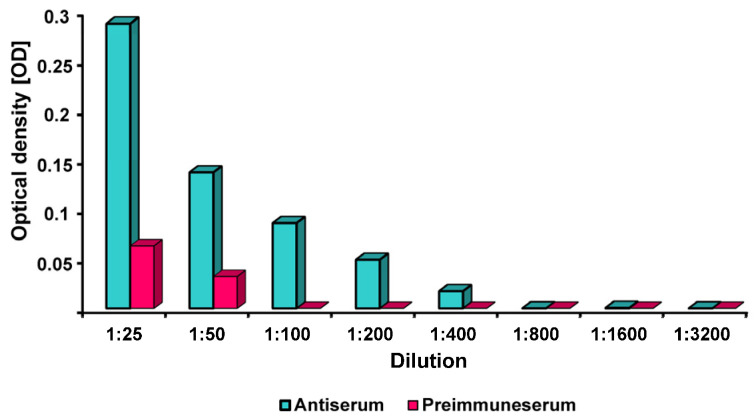
Titre (=dilution) of the FimA rabbit antiserum compared to the preimmune serum by optical density (OD).

**Figure 2 ijms-26-04726-f002:**
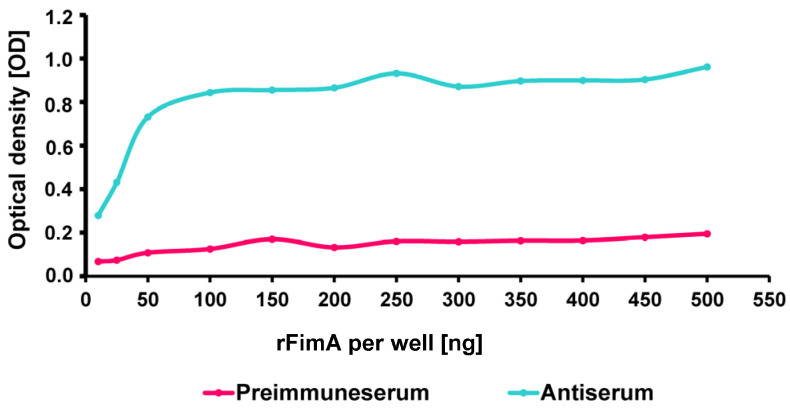
Determination of the best coating concentration of the recombinant FimA (ng per well) by optical density (OD).

**Figure 3 ijms-26-04726-f003:**
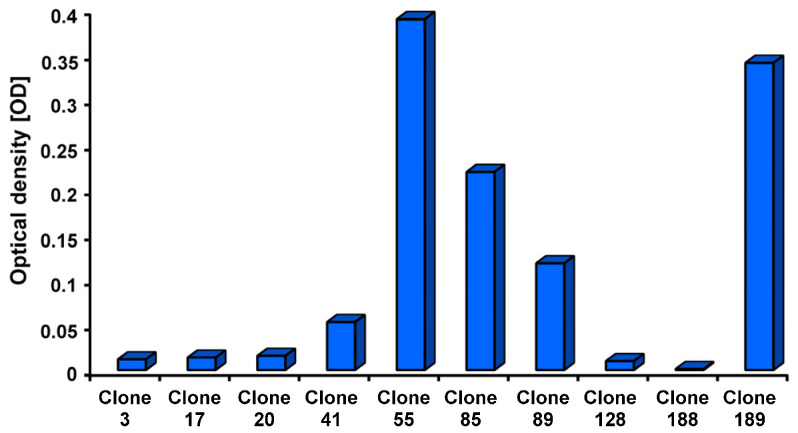
The optical densities (ODs) of the tested mouse monoclonal antibodies of 10 clones (*n* = 3).

**Figure 4 ijms-26-04726-f004:**
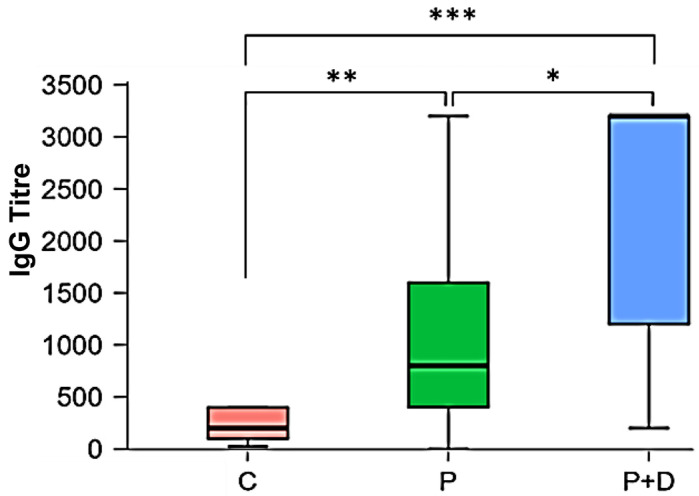
Comparison of the IgG titres of the 13 controls (C), the 26 periodontitis patients (P) and the 15 patients with periodontitis and diabetes mellitus type 2 (P + D) (*n* = 3); * *p* < 0.05, ** *p* < 0.01, *** *p* < 0.001.

**Figure 5 ijms-26-04726-f005:**
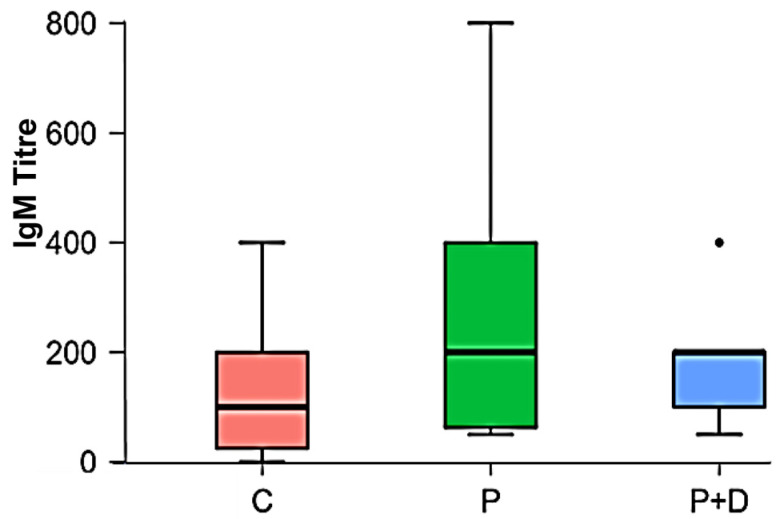
Comparison of the IgM titres of the 13 controls (C), the 26 periodontitis patients (P) and 15 patients with periodontitis and diabetes mellitus type 2 (P + D) (*n* = 3); the dot in P + D marks one outlier.

**Figure 6 ijms-26-04726-f006:**
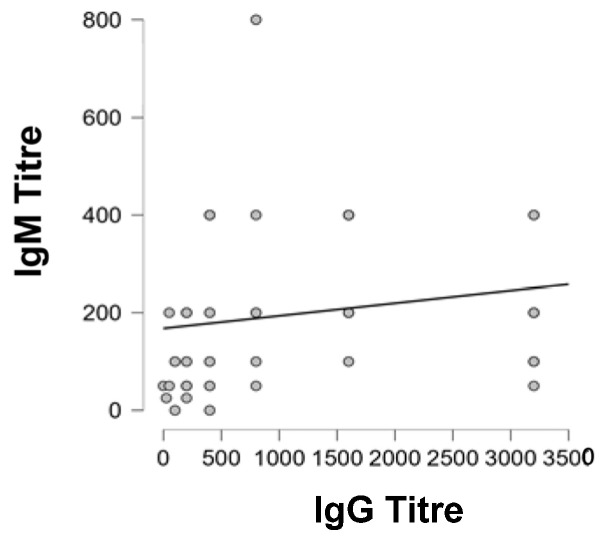
Spearman’s rank correlation showing the relationship between the IgG titres and the IgM titres.

**Table 1 ijms-26-04726-t001:** Demographic data of the patients. C: healthy individuals; P: periodontitis patients; P + D: patients with periodontitis and diabetes mellitus type 2; SD: standard deviation.

Cohort	Number	Female	Male	Mean Age (years)	SD Age [years]	Variance Age [years^2^]
C	13	5	8	44	9	86.3
P	26	14	12	51	12	164.8
P + D	15	7	8	47	9	93.6

**Table 2 ijms-26-04726-t002:** Descriptive statistic of the IgG titres of the cohorts. C: orally healthy individuals; P: periodontitis patients; P + D: patients with periodontitis and diabetes mellitus type 2.

Descriptive Statistics: IgG Titres			
Diagnosis	C	P	P + D
Number	13	26	15
Mode	1:200; 1:400	1:1600	1:3200
Median	1:200	1:800	1:3200
Shapiro–Wilk	0.848	0.825	0.762
*p*-value Shapiro–Wilk	0.027	<0.001	0.001

**Table 3 ijms-26-04726-t003:** Absolute and relative frequency distribution of the IgG titres. C: orally healthy individuals; P: periodontitis patients; P + D: patients with periodontitis and diabetes mellitus type 2.

		IgG Titres									
Diagnosis		No Titre	1:25	1:50	1:100	1:200	1:400	1:800	1:1600	1:3200	Total
C	Number	0	1	1	3	4	4	0	0	0	13
	%	0	7.69	7.69	23.08	30.77	30.77	0	0	0	100
P	Number	1	0	1	0	4	6	5	7	2	26
	%	3.85	0	3.85	0	15.39	23.08	19.23	26.92	7.69	100
P + D	Number	0	0	0	0	2	1	1	3	8	15
	%	0	0	0	0	13.33	6.67	6.67	20	53.33	100

**Table 4 ijms-26-04726-t004:** Kruskal–Wallis test and Levene test of the IgG titres; df: number of degrees of freedom; *p*: *p*-value; *** *p* < 0.001.

Kruskal–Wallis Test			Levine Test	
Factor	Statistics	df	*p*	*p*
diagnosis	20.951	2	<0.001 ***	<0.001 ***

**Table 5 ijms-26-04726-t005:** Dunn’s post hoc test with Bonferroni–Holm correction of the cohorts IgG titres. C: orally healthy individuals; P: periodontitis patients; P + D: patients with periodontitis and diabetes mellitus type 2. * *p* < 0.05, ** *p* < 0.01, *** *p* < 0.001.

Dunn’s Post Hoc Test with Bonferroni–Holm Correction	
Comparison	*p*
C − P	0.008 **
C − P + D	<0.001 ***
P − P + D	0.02 *

**Table 6 ijms-26-04726-t006:** Descriptive statistics of the IgM titres of the cohorts. C: orally healthy individuals; P: periodontitis patients; P + D: patients with periodontitis and diabetes mellitus type 2.

Descriptive Statistics: IgM Titres			
Diagnosis	C	P	P + D
Number	13	26	15
Mode	1:200	1:50; 1:400	1:200
Median	1:100	1:200	1:200
Shapiro–Wilk	0.862	0.803	0.819
*p*-value Shapiro–Wilk	0.041	<0.001	0.007

**Table 7 ijms-26-04726-t007:** Absolute and relative frequency distribution of the IgM titres. C: orally healthy individuals; P: periodontitis patients; P + D: patients with periodontitis and diabetes mellitus type 2.

Kruskal–Wallis Test			
Factor	Statistics	df	*p*
Diagnosis	2.504	2	0.286

**Table 8 ijms-26-04726-t008:** Spearman’s correlation of IgG to IgM titres; ** *p* < 0.01.

Spearman’s Correlations			
Comparison		Spearman’s Rho	*p*
IgG Titre	IgM Titre	0.421	0.002 **

**Table 9 ijms-26-04726-t009:** Inclusion/exclusion criteria for the chosen sera of patients/controls.

Inclusion Criteria			Exclusion Criteria
Periodontitis Patients	Periodontitis Plus Diabetes Mellitus Type 2 Patients	Orally Healthy Controls	Whole Cohort
Chronic periodontitis: periodontal screening index (PSI) ≥ 4	Chronic periodontitis: periodontal screening index (PSI) ≥ 4	Orally healthy individuals: periodontal screening index (PSI) < 1	Pregnancy
			Smoking
			BMI < 18.5 kg/m^2^
			Persistent alcohol, drug or medication abuse
Age 18–70 years	Diabetes mellitus type 2 with HbA1c-value ≥ 8.5% Age 18–70 years	Age 18–70 years	Severe cardiovascular diseases (coronary heart disease, cerebral vascular disease, peripheral vascular disease, heart valve disease, heart failure)
All genders	All genders	All genders	Allergies to supportive medications/antiseptics and dental materials (e.g., gloves or chlorhexidine)
Patients with at least 12 natural teeth (without subgingival inlay, tooth crown or caries)	Patients with at least 12 natural teeth (without subgingival inlay, tooth crown or caries)	Individuals with at least 12 natural teeth (without subgingival inlay, tooth crown or caries)	Other serious diseases (including cancer, liver diseases, lung diseases, chronic infectious diseases like HIV, hepatitis, rheumatological diseases, haematological diseases, severe psychiatric illness
			Systemic enteral or parenteral medication such as daily vitamins or nutritional supplements and certain calcium canal blockers (e.g., nifedipine) besides antidiabetic/insulin replacement
			Allergies to supportive medications/antiseptics and dental materials (e.g., gloves or chlorhexidine)
			Serious dental diseases (severe caries and/or pulp diseases requiring extensive dental, surgical or prosthetic treatment, diseases requiring systemic medication, systemic, topical or inhaled steroid treatment for more than 30 consecutive days within 6 weeks before the initial examination)
			Any periodontal treatment within 6 months prior to study entry

## Data Availability

The raw data supporting the conclusions of this article will be made available by the authors on request.

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
