# Peer review of "Serum Antibody Titres Against Porphyromonas gingivalis Fim A in Patients with Periodontitis with and Without Diabetes Mellitus Type 2"

_ijms, 2025, doi:10.3390/ijms26104726_

Round 1
Reviewer 1 Report
Comments and Suggestions for Authors
L63: can't --> cannot.
L86: "differentiation 2between". Remove 2.
Table 4: Gesamt --> Total.
Overall, the manuscript is easy to follow, and shows novelty. The Discussion can be further strengthened. For example, the Conclusion says: "a reliable and valid ELISA test for the detection of antibody titres against P. gingivalis FimA could be established, which holds the potential to serve as useful test for a possible existence of undiagnosed periodontitis w/o diabetes." Patients should go to see a dentist to diagnose periodontitis. Who will do an ELISA to check IgG against P. Gingivalis FimA? I don't think medical doctors will request a lab test for diabetes patients or any other medical patients. This is too specific. Or, if this ELISA test can be self-administered at home, then dental phobia individuals may choose to buy this over-the-counter kit and check at home. Currently, it sounds a bit far-fetched. But the Discussion section can be enriched by having a conversation about the future direction or potential clinical relevance of this piece of work.
Author Response
Reviewer 1 Comments
L63: can't --> cannot.
Response: This was performed as requested.
L86: "differentiation 2between". Remove 2.
Response: This was performed as requested.
Table 4: Gesamt --> Total.
Response: This was performed as requested.
Overall, the manuscript is easy to follow, and shows novelty.
Response: We thank the reviewer for this positive appraisal.
The Discussion can be further strengthened. For example, the Conclusion says: "a reliable and valid ELISA test for the detection of antibody titres against P. gingivalis FimA could be established, which holds the potential to serve as useful test for a possible existence of undiagnosed periodontitis w/o diabetes." Patients should go to see a dentist to diagnose periodontitis. Who will do an ELISA to check IgG against P. Gingivalis FimA? I don't think medical doctors will request a lab test for diabetes patients or any other medical patients. This is too specific. Or, if this ELISA test can be self-administered at home, then dental phobia individuals may choose to buy this over-the-counter kit and check at home. Currently, it sounds a bit far-fetched. But the Discussion section can be enriched by having a conversation about the future direction or potential clinical relevance of this piece of work.
Response: We thank the reviewer for these helpful suggestions. The proposed conversation is included in the discussion now and marked in yellow.
Reviewer 2 Report
Comments and Suggestions for Authors
Serum antibody titres against Porphyromonas gingivalis Fim A 2 in patients with periodontitis with and without diabetes mellitus type 2
This is an interesting study for clinicians and practitioners since it addresses the problems of periodontitis-diabetes correlations.
Introduction provides all the necessary information for the reader to understand the subject.
Methods are clear and reproducible. The inclusion and exclusion criteria are present, as well as data regarding the statistical analysis.
Results are clear and in line with objectives.
Discussion: The necessary correlation with current research flow is present.
Conclusion. In my opinion, the “in summary” could be converted into conclusions.
Author Response
Reviewer 2 Comments
This is an interesting study for clinicians and practitioners since it addresses the problems of periodontitis-diabetes correlations.
Introduction provides all the necessary information for the reader to understand the subject.
Methods are clear and reproducible. The inclusion and exclusion criteria are present, as well as data regarding the statistical analysis.
Results are clear and in line with objectives.
Discussion: The necessary correlation with current research flow is present.
Response: We thank the reviewer for this positive and encouraging appraisal.
Conclusion. In my opinion, the “in summary” could be converted into conclusions.
Response: This was performed as requested.